# Unsupervised open-vocabulary action recognition with an autoregressive model

## Abstract

Current works on zero/few- shot action recognition are largely based on contrastive approaches trained in a supervised manner to select an action class out of a predefined set. Instead, in this work, we propose a new paradigm for zero-shot action recognition based on autoregressive generation of a free-form action-specific caption describing the action occurring in the video. To this end, we propose to adapt an image-based pre-trained *autoregressive* Vision & Language (V&L) Model for action recognition. We firstly show that direct fine-tuning of an autoregressive model using the action classes suffers from severe overfitting. To alleviate this, we then introduce an unsupervised learning framework consisting of two key components: (a) an unsupervised method for adapting the autoregressive model to action/video data by means of pseudo-caption generation and *self-training* without using any action-specific labels; (b) a *retrieval* component for discovering a diverse set of pseudo-captions for each video. In the process, we show that both components are necessary to obtain high accuracy. Our model results in predictions that are fine-grained, interpretable, and naturally open-vocabulary. Importantly, when evaluated for zero- and few-shot action recognition, our approach matches or even outperforms contrastive learning-based methods.

## 1 Introduction

This work is on adapting a V&L FM (Radford et al., 2021; Jia et al., 2021; Yuan et al., 2021; Yu et al., 2022; Alayrac et al., 2022; Li et al., 2022; Wang et al., 2022) for video/action recognition. In contrast to all prior work that use a V&L model trained with contrastive learning, i.e. CLIP (Wang et al., 2021; Ju et al., 2022; Castro & Heilbron, 2022; Lin et al., 2022c), our focus is on adapting an *autoregressive* V&L model, and specifically BLIP (Li et al., 2022), to the video domain with the goal of producing an action-specific caption in an autoregressive manner. Besides this being a challenging research goal on its own, an autoregressive approach has practical advantages including producing more fine-grained (compared to an action class label) and more interpretable/human-readable output, and being naturally more open-world (compared to contrastive learning based approaches) in a sense that it does not require a priori definition of the action classes.

In prior work (Wang et al., 2021; Ju et al., 2022; Castro & Heilbron, 2022; Lin et al., 2022c), CLIP adaptation to action/video recognition was seamlessly done using the standard contrastive loss and hand-engineered or learned prompts encoding action class information. In contrast, direct end-to-end fine-tuning of an autoregressive model on class names results in severe base class overfitting and poor open-world zero-shot generalizability (see Table 1). Zero-shot generalization is a fundamental property of V&L models that we wish not to sacrifice when adapting the model to the video domain.

To alleviate the aforementioned overfitting problem, we propose to entirely drop action labels during training and adopt an unsupervised adaptation approach. We start from an image-based autoregressive model (i.e. BLIP (Li et al., 2022)). We describe a self-training procedure where the model is iteratively trained on video data, without any labels, using an autoregressive objective where the training pseudo-captions are produced by the model itself. In this case, the model is able to generalize much better to unseen classes and, at the same time, its label-free nature makes it suitable for any large-scale video dataset. However, our findings suggest that **self-training solely is not sufficient to train a highly accurate model** as the generated pseudo-captions are lacking diversity [1].

---

[1] Some sort of the so-called confirmation bias (Arazo et al., 2020).

Table 1: In this work, we propose to train an Autoregressive Action Recognition (AAR) model for open-vocabulary action recognition. The table shows zero-shot generalization on Kinetics-220 (1-vs-620 setting; see Sec. 6 for more details) of AAR trained in 3 different ways: (a) action class labels, (b) pseudo-captions, and (c) pseudo-captions + retrieval (as proposed in this work). Training with class labels results in very poor zero-shot generalization. Self-training with pseudo-captions helps, but crucially high accuracy can be achieved only when the proposed retrieval mechanism is integrated within self-training.

| Method | Supervisory signal | Retrieval | Top-1 | Top-5 |
|---|---|---|---|---|
| Fully supervised | Class names | ✗ | $0.76 \pm 0.4$ | $37.8 \pm 1.0$ |
| Self-training | Pseudo-captions | ✗ | $14.70 \pm 0.85$ | $44.85 \pm 0.91$ |
| **REtrieval & Self-Training** | Pseudo-captions | ✓ | $\mathbf{29.51 \pm 0.71}$ | $\mathbf{56.12 \pm 0.37}$ |

To alleviate this, we **further propose a retrieval mechanism, integrated into self-training**, which is tasked with finding, for each training video, additional pseudo-captions from semantically-similar videos. These pseudo-captions are then used to enhance learning by means of increasing the diversity of the training data. Critically, we show that our retrieval approach drastically increases the accuracy and the quality of the trained models. In summary, our **main contributions** are:

- To the best of our knowledge, we propose the **very first method** for adapting an **autoregressive** V&L FM for **open-vocabulary action/video recognition**.

- We introduce RetrIeve & SElf-Train (RISE), a training framework consisting of two key components; (a) iterative **Self-training without** using any action-specific **labels**, and (b) CLIP-based **Retrieval** for discovering a diverse set of pseudo-captions to train the model. We show that **both components are necessary** to train a high quality model.

- We evaluate our approach on zero-shot action recognition, where we show that our approach is very competitive with respect to CLIP-based methods, matching and/or surpassing the state-of-the-art on standard evaluation benchmarks.

## 2 RELATED WORK

**Unsupervised & Semi-supervised Image Captioning:** There are only very few methods that have attempted to train an image captioning model without full supervision. (Chen et al., 2017) propose a method that transfers a COCO model to other domains by means of adversarial training using unpaired data in the target domain. Similar in spirit, approaches focusing on training an LSTM discriminator to distinguish real from generated captions were proposed in (Feng et al., 2019; Laina et al., 2019; Zhou et al., 2021). Moreover, semi-supervised approaches include Kim et al. (2019) which combines paired and unpaired data with adversarial training, and Chen et al. (2021) which performs iterative self-training using a mean teacher consisting of an ensemble of independently trained models. Compared to Chen et al. (2021), which is the closest work to ours from the above, our method (a) does not use a mean teacher but critically a *retrieval* component which is shown to greatly improve the generated captions, and (b) focuses on image-to-action/video captioning (rather than on image-to-image) which is significantly more challenging.

**V&L Foundation Models for Action/Video Recognition:** Following the development of CLIP, a number of very recent works have attempted to adapt it to the video domain. X-CLIP (Lin et al., 2022c) trains a lightweight transformer on top of CLIP image features for spatio-temporal fusion. Ju et al. (2022) uses soft prompt learning to adapt CLIP to the video domain, while Wang et al. (2021) performs standard end-to-end finetuning. Starting from CLIP, FitCLIP (Castro & Heilbron, 2022) proposes a teacher-student approach based on a small video-text labelled dataset and pseudo-labels generated on a large unlabeled dataset. Notably, their method uses an ensemble, largely relying on the original CLIP model to produce high accuracy. The aforementioned works use contrastive learning and labelled data for CLIP-to-video adaptation, in a fully-supervised fashion. In contrast, our work is the first, to our knowledge, to adapt an autoregressive V&L model to the action/video domain. While this is a significant challenge on its own, notably we do so without using any labels.

**Video Captioning:** A large body of methods are trained in a fully supervised manner using caption annotations (which are costly), see (Pan et al., 2020; Tang et al., 2021b;a; Lin et al., 2022b; Liu et al., 2022). In contrast, RISE does not use any human annotations to train the models.

**Zero-shot Action Recognition:** Most works on zero-shot action recognition build on learning attribute-based *semantic embeddings* from video features in order to make them be close to the word embedding of the class names (Zhu et al., 2018; Brattoli et al., 2020; Bretti & Mettes, 2021; Mettes, 2022; Estevam et al., 2022; Pu et al., 2022; Luo et al., 2022), or use a text encoder which is learned or updated as part of the training (Ni et al., 2022; Ge et al., 2022; Qian et al., 2022; Lin et al., 2022a; Qian et al., 2022). Most recent approaches using word embeddings focus on the *alignment* problem i.e. learning visual representations that match the corresponding class semantic embeddings but also generalize to unseen class embeddings (Pu et al., 2022; Luo et al., 2022). All the aforementioned methods above operate in a discriminative setting where the class embeddings need to be computed. In contrast, our method is generative, being able to generate action-specific captions in an autoregressive manner without having to manually pre-define the classes of interest.

## 3 METHOD

### 3.1 OVERVIEW

We are given an action recognition dataset $D$ consisting of $N$ video clips $v_i$, $i = 1, \ldots, N$ *without the class labels*. We construct an autoregressive action recognition model consisting of a video encoder $g_v(.)$ and an autoregressive text decoder $g_t(.)$. The model is instantiated from a pre-trained autoregressive V&L model, i.e., BLIP (Li et al., 2022). Note, however, that our framework is agnostic to the exact architecture of the autoregressive model. For training, we use a standard language modeling loss. In the proposed framework, we iteratively train AAR on pseudo-captions produced by the model itself (self-training). Crucially, we describe how this process is greatly enhanced by increasing the diversity of the pseudo-captions by using a retrieval module based on CLIP. We coin our approach: **R**etr**I**eve & **SE**lf-Train (RISE).

The AAR model and the language modeling loss to train it are described in Sec. 3.2 and Sec. 3.3, respectively. The retrieval module is described in Sec. 3.4. Finally, the integrated retrieve & self-train framework is described in Sec. 3.5. The overall approach is depicted in Fig. 1.

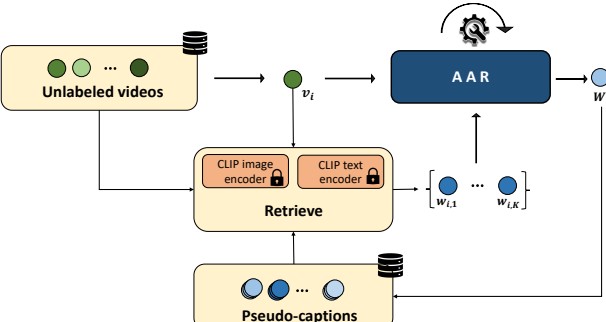

Figure 1: **RISE overview.** Our goal is to train an Autoregressive Action Recognition (AAR) model without labels. In the proposed framework, we iteratively train AAR on pseudo-captions produced by the model itself (self-training) using a retrieval mechanism. For a given input video $v_i$, the retrieval mechanism finds in the image space of CLIP the $H$ closest videos, and then, using the CLIP text encoder, selects the $K$ pseudo-captions that explain the video better. Finally, the set of pseudo-captions for the video dataset are updated using newly refined predictions from AAR.

### 3.2 AUTOREGRESSIVE ACTION RECOGNITION (AAR) MODEL

Both the video encoder and the autoregressive text decoder are instantiated from an image-text autoregressive model. Unless otherwise specified, we use a BLIP model (Li et al., 2022). We largely maintain BLIP's architecture by only modifying its visual transformer to combine temporal information across frames using a temporal adapter. See appendix for a graphical depiction.

The **text decoder** $g_t(.)$ consists of 12 transformer layers, each performing causal attention on the text tokens and cross-attention between the text and the visual tokens produced by the video encoder.

The **video encoder** $g_v(.)$ is constructed by inserting a temporal adapter after each self-attention layer of BLIP's spatial visual transformer (a ViT (Dosovitskiy et al., 2021)). The visual transformer layers continue to operate independently on each frame (i.e. spatial attention only) while the adapter will pool and combine information across frames, handling the temporal modeling aspect. Specifically, let $\mathbf{Z}^l \in \mathbb{R}^{T \times H \times W \times d}$ be the output of the spatial transformer's layer $l$ (the class token $\mathbf{Z}_{cls}^l$ is excluded). The adapter operates on $\mathbf{Z}^l$ as follows:

$$\mathbf{Z'}^l = \mathbf{Z}^l + a_d(\mathbf{Z}^l), \tag{1}$$

where $a_d(.)$ takes the form of a $3 \times 3 \times 3$ 3D depth-wise convolution. The adapter layer is initialized such that the temporal residual, introduced by it, will be of low magnitude initially, closely aligning the network's output early on to that of the pre-trained image model. This helps stabilize the training during the initial iterations. Finally, $\mathbf{Z'}^l$, along with the class tokens $\mathbf{Z}_{cls}^l$, are averaged over $T$, to compute video feature $\mathbf{F}^l \in \mathbb{R}^{(H \cdot W + 1) \times d}$ that interacts with the text decoder.

### 3.3 LANGUAGE MODELLING LOSS

The language modelling objective $\mathcal{L}_{LM}$ is the negative log-likelihood of the self-generated pseudo-captions, and it is used as the main loss to update our model $g_v(.)$ and $g_t(.)$ To compute the loss, we compute the visual tokens $\mathbf{F} \in \mathbb{R}^{(H \times W + 1) \times d} = g_v(\mathbf{X})$ from the video frames $\mathbf{X}$, and we are given a pseudo-caption $\tilde{w}$. As we shall see in Sec. 3.5, $\tilde{w}$ is sampled from a set of cached pseudo-captions initially generated by BLIP and then updated by $g_t(.)$ over the course of training. We compute the text tokens as $y_{\tilde{w}} = \phi(\tilde{w}) \in \mathbb{N}^M$, with $\phi$ being a tokenizer function that maps (sub-)words into the one-hot vectors spanning the vocabulary size. Following standard practices in image captioning, we pre-pend $y_{\tilde{w}}$ with a $y_0 = [\text{BOS}]$ token as well as a prompt $y_p = \phi(\text{"A video of "}) \in \mathbb{N}^P$. The input to the model's decoder $g_t(.)$ is set as $y = [y_0, y_p, y_{\tilde{w}}] \in \mathbb{N}^{1+P+M}$. The text decoder applies left-to-right masked attention (i.e. causal attention) and produces an output $o = [o_i]_{i=1}^{1+P+M}$ with $o_i = g_t(y_{i' < i} | \mathbf{F})$. The language modelling loss is then computed using standard cross-entropy (CE):

$$\mathcal{L}_{LM}(y_{\tilde{w}}) = \sum_{i=1+P}^{1+P+M} CE(y_i, o_i). \tag{2}$$

### 3.4 VIDEO-VIDEO & VIDEO-TEXT RETRIEVAL

A key component of RISE is the integration of a retrieval mechanism into the self-training process. Our framework uses video-video and video-text retrieval modules both instantiated from a pretrained CLIP model (Radford et al., 2021) that remains frozen over the training process. We selected CLIP due to its accuracy and strong generalizability properties (Radford et al., 2021).

Given a video $\mathbf{X}$ and pseudo-caption $w$, we use the CLIP image $g_I^C(.)$ and text $g_T^C(.)$ encoders to compute video $\mathbf{f}_C = \sum_t g_I^C(\mathbf{X}_t) \in \mathbb{R}^d$ and text $\mathbf{t}_C = g_T^C(w) \in \mathbb{R}^d$ features, respectively. For a given pair of videos $i, j$, and a given pair of video-caption $i, j$, the video-video $s_{vv}$ and video-text $s_{vt}$ similarities are computed as:

$$s_{vv}^{i,j} = \mathbf{f}_C^i \cdot \mathbf{f}_C^j \quad (3) \qquad s_{vt}^{i,j} = \mathbf{f}_C^i \cdot \mathbf{t}_C^j, \quad (4)$$

where we use the subscript $C$ to refer to features computed by the frozen CLIP model.

### 3.5 RETRIEVE & SELF-TRAIN (RISE)

This section describes our framework for training the model of Sec. 3.2 without using any human-annotated action classes/captions. Instead, the model is trained on pseudo-captions generated by the model itself in a self-training manner. We also introduce a retrieval module based on CLIP for increasing the diversity of the pseudo-captions used to supervise the training for each input video, which is shown to greatly enhance the learning process. RISE is summarized in Algorithm 1. **Self-train:** The process is iterative and, at each training iteration, video $v_i$ maintains a list of $K$ associated pseudo-captions $\tilde{W}_i = [\tilde{w}_{i,k}]$, $k = 1, \ldots, K$ describing its content. From this list, a pseudo-caption is randomly sampled and used as supervisory signal for $v_i$ to train the model using Eq. 2. After training for $R$ epochs, the model produces a new pseudo-caption for $v_i$ denoted as $\tilde{w}_{i,new}$. This is added to the existing list, resulting in $K + 1$ pseudo-captions $\tilde{W}_i \leftarrow \tilde{W}_i \cup \tilde{w}_{i,new}$.

---

**Algorithm 1** RISE Training

---

**Require:** $\{v_i\}$, $i = 1 : N$ clips, pre-trained CLIP and BLIP models.
 1: Compute $\tilde{W}_i = [\tilde{w}_{i;t} = g_v^{BLIP}(\mathbf{X}_{i;t})]$          $\triangleright\ t = 1, \ldots, T\ ,\ i = 1, \ldots, N$
 2: Compute $\mathbf{f}_C^i, \mathbf{t}_C^i = g_{I,T}^C(\mathbf{X}_i; \tilde{W}_i)$ (Sec. 3.4)        $\triangleright\ i = 1, \ldots, N$
 3: Compute $s_{vv}^{i,j}$ (Eq. 3) and $s_{vt}^{i,j}$ (Eq. 4)      $\triangleright\ i = 1, \ldots, N,\ j = 1, \ldots, N$
 4: Update $\tilde{W}_i$ (Eqs. 5-6)     $\triangleright$ Retrieve top-$K$ captions from top-$H$ similar videos
 5: **while** training **do**
 6:      **for** $R$ epochs **do**
 7:          Sample batch $v_i$ and $\tilde{w}_i \in \tilde{W}_i$
 8:          Update $g_t$ and $g_v$ using Eq. 2
 9:      **end for**
10:      Compute $\tilde{W}_i \leftarrow \tilde{W}_i \cup [\tilde{w}_{i;new} = g_t(g_v(\mathbf{X}_i))]$      $\triangleright\ i = 1, \ldots, N$
11:      Update $\mathbf{t}_C^i = g_T^C(\tilde{W}_i)$           $\triangleright\ i = 1, \ldots, N$
12:      Compute $s_{vv}^{i,j}$ (Eq. 3) and $s_{vt}^{i,j}$ (Eq. 4)    $\triangleright\ i = 1, \ldots, N,\ j = 1, \ldots, N$
13:      Update $\tilde{W}_i$ (Eqs. 5-6)     $\triangleright$ Retrieve top-$K$ captions from top-$H$ similar videos
14: **end while**

---

**Retrieve:** The retrieval module is used to update the pseudo-caption list for each video $v_i$ over the course of self-training. Specifically, for each $v_i$, we use the video-video similarity of Eq. 3 to compute $\mathbf{s}_{vv}^i = s_{vv}^{i,j}$, $j = 1, \ldots, N$, and then retrieve the corresponding $K + 1$ captions associated to the $H$ most similar videos to $v_i$, creating a large list of $H(K+1)$ captions as:

$$\Omega_i = \bigcup_{j \in \text{top-}H} \tilde{W}_j \tag{5}$$

We then compute the CLIP-based text embeddings $\mathbf{t}_C^j = g_T^C(\tilde{w}_j)$ for $\tilde{w}_j \in \Omega_i$, and the $H(K+1)$ video-text similarity scores $s_{vt}^{i,j} = \mathbf{f}_C^i \cdot \mathbf{t}_C^j$. Finally, we update $\tilde{W}_i$ for video $v_i$ by keeping the top-$K$ most relevant captions from $\tilde{W}_i \cup \Omega_i$, i.e. we update the list of captions $\tilde{W}_i$ for video $v_i$ as:

$$\tilde{W}_i \leftarrow \bigcup_{i \in \text{top-}K} \left[ \tilde{w}_i \in \left[ \tilde{W}_i \cup \Omega_i \right] \right]. \tag{6}$$

**Initialization:** At the beginning of the training process, the videos $v_i$, $i = 1, \ldots, N$ are populated with pseudo-captions by an off-the-shelf captioning model. To this end, we use BLIP's text decoder to produce a caption $\tilde{w}_{i,t}$ for a set of $T$ video frames $t = 1, \ldots, T$. To encourage the generation of action-specific outputs, we use a set of manual prompts, such as "`a video of`", "`a person is`" or "`someone is`", to initialize the text decoder's output.

**Training efficiency:** Besides training the model with the language modelling loss, the above self-training framework includes a retrieval step, which can potentially render the training process slow. However, note that the video-video scores $\mathbf{s}_{vv}^i$ for each video $v_i$ are computed using a frozen CLIP model, and, hence, all these scores can be pre-computed and re-used over the course of training. Moreover, the video feature used to compute the video-text scores $\mathbf{s}_{vt}^i$ can also be pre-computed for all videos, and only the text features corresponding to newly produced captions need be computed during training. Hence, $\mathbf{s}_{vt}^i$ can be also efficiently computed during training.

## 4   EVALUATING AUTOREGRESSIVE ACTION RECOGNITION

While evaluating standard classification models is trivial, this is not the case for our autoregressive action recognition model, given that its output is free unconstrained text. Direct character-by-character assessment is complicated as (a) there is more than one action caption that could describe the video, and (b) the same action can be expressed in multiple ways. Assessing the quality of generated text is an open research question, and various metrics such as CIDEr (Vedantam et al., 2015), BLEU (Papineni et al., 2002) and ROUGE (Lin, 2004) have been proposed to evaluate the correctness of the generated captions from the perspective of human judgment. However, such metrics are hard to correlate with the typical accuracy score expected in a classification problem and tend to penalize predictions outside the expected vocabulary disproportionately.

To alleviate this, we propose to capitalize on the capacity of the CLIP text encoder to produce semantically distinctive features, ignoring elements of grammar completeness such as pronouns or adverbs, and being also invariant to permutations in the position of the class names. Given a set of class names $\mathcal{C} = \{\gamma_1, \ldots \gamma_{|\mathcal{C}|}\}$, we compute the class embeddings using the CLIP text encoder $\mathbf{w}_c = g_T^C(\gamma_c)$, compute the CLIP embedding for a caption $\tilde{w}$ generated by our model as $\mathbf{t}_C = g_T^C(\tilde{w})$, and select the target class from $\mathcal{C}$ as $\hat{c} = \arg\max_c \mathbf{w}_c \cdot \mathbf{t}_C$. By associating a class to the predicted caption, we can directly compute an accuracy score, reducing the problem to standard top-k closed-set classification. We call this metric CLIP-based top-k emphasizing that it is directly comparable with standard top-k.

**User study:** We evaluate the capacity of CLIP-based top-k to correlate with human judgment by conducting a user study on 1,000 videos randomly selected from Kinetics-600. The study concluded that human annotators agree with the CLIP-based top-k metric in 75.67% of the cases, further validating the correctness of the latter to act as a proxy for classification. See appendix for details.

## 5 EXPERIMENTAL SETTINGS

Following the standard zero-shot action recognition evaluation setting (e.g. (Ni et al., 2022; Chen & Huang, 2021; Pu et al., 2022)), our models are trained on the training set of Kinetics-400 (Kay et al., 2017) and tested on HMDB-51 (Kuehne et al., 2011), UCF-101 (Soomro et al., 2012) and Kinetics-600 (Carreira et al., 2018) (the splits from (Chen & Huang, 2021)). Note however, that RISE training is performed *without using any labels*.

For all the experiments, unless otherwise stated, we sample uniformly 8 frames at a res. of $224 \times 224$px. During inference, we follow best practices (Li et al., 2022; Wang et al., 2022) and use beam search to generate the captions. To showcase the generalizability of our approach we report results for two AR architectures: BLIP (Li et al., 2022) and GIT (Wang et al., 2022). We will refer to them simply as RISE for the variant based on BLIP and RISE (GIT) for the one based on GIT.

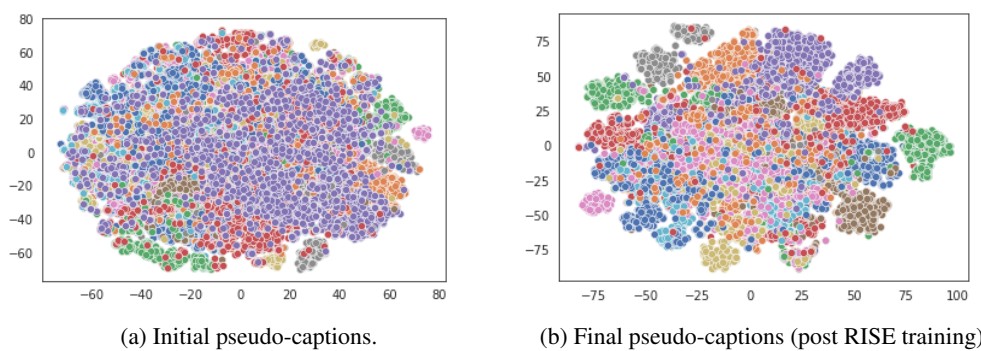

(a) Initial pseudo-captions.  (b) Final pseudo-captions (post RISE training).

Figure 2: **t-SNE plot** (in the CLIP text embedding space) showcasing the feature distribution before and after RISE. For clarity, we only plot, in both cases, 25 random classes. RISE significantly improves the quality of the clusters.

**Zero-shot experiments**: We train our model following Algorithm 1 on the $\sim 200K$ videos comprising the training set of Kinetics-400, without using any labels. We initialize the algorithm with the following parameters: the number of nearest neighbors $H$ is set to 2,000, the size of the pseudo-caption cache per video is $K = 3$, and the number of epochs $R$ between retrievals is set to 10. The model is trained for 60 epochs. The algorithm and the models were implemented using Py-Torch (Paszke et al., 2019). The full list of hyperparameters is reported in the appendix. We evaluate the accuracy of our method using our CLIP-based top-k (directly comparable with the top-k accuracy used in prior works; see also Sec. 4).

*Evaluation protocol*: On HMDB-51 and UCF-101, following Ni et al. (2022), we report the average top-1 accuracy and standard deviation computed across each of the three test splits. Similarly, on Kinetics-600, we perform evaluations on the three testing splits introduced in Chen & Huang (2021). Each split contains videos belonging to 160 classes different from the ones found in Kinetics-400, with a total of 220 unique classes among the three splits. Note that Chen & Huang (2021) reas-

signed the labels to ensure that there is no overlap between the defined 220 classes and the 400 ones from Kinetics-400. Further to this standard setting, we also report results for the more challenging generalized zero-shot setting.

**Few-shot experiments:** We also validate our model for few-shot recognition, where we finetune the trained models directly using the class names (instead of action-specific captions). Depending on the number of samples per class available at train time, we finetune the model between 50 (for 16-shot) and 200 epochs (for 2-shot) using the same hyperparameters used for the RISE training.

*Evaluation protocol*: We apply the standard $M$-shot setting with $M = \{2, 4, 8, 16\}$ training videos per class. We report the average Top-1 and Top-5 accuracy on UCF-101 and HMDB-51 after training and evaluating on each of the 3 train-test splits.

# 6 RESULTS

## 6.1 ABLATION STUDIES

Unless otherwise stated, all of our ablations are performed on HMDB-51 dataset using BLIP as the AAR architecture for RISE.

**Evaluation against vanilla BLIP & GIT baselines:** In Table 2, we measure the gains offered by our method against directly using the image captioning baselines, BLIP and GIT, for zero-shot recognition. As the results show, for both architectures, RISE significantly improves upon the baseline results provided by BLIP and GIT.

**Effect of retrieval in RISE:** A key component of RISE is integrating retrieval into self-training. Without the retrieval component, the model simply reinforces its own predictions and is unable to fix its mistakes (a form of self-confirmation bias). We analyze its importance by conducting an experiment where only self-training is used (i.e. $H = 1$). We see that the results on HMDB-51 drop massively from $49.7\%$ to $40.1\%$ (see Table 3), clearly validating that **the retrieval step in RISE is crucial for training a high quality model.**

Table 2: Comparison between RISE, BLIP (Li et al., 2022) and GIT (Wang et al., 2022) baselines for zero-shot recognition on HMDB-51. *CC12+CC2M+SBU.

| Method | Dataset | Top-1 |
|---|---|---|
| BLIP | LAION-115M | 30.3 |
| RISE (Ours) | Kinetics-400 | **49.7** |
| GIT | 14M* | **34.6** |
| RISE (GIT) (Ours) | Kinetics-400 | 50.1 |

Table 4: Ablation studies on HMDB-51.

| | K = 1 | K = 3 | K = 5 |
|---|---|---|---|
| Ours | 48.4 | **49.7** | 49.7 |

(a) Effect of number of cached captions.

| $N_I = 1$ | $N_I = 2$ | $N_I = 3$ | $N_I = 4$ |
|---|---|---|---|
| 46.0 | 48.8 | **49.7** | 49.8 |

(b) Effect of number of retrieval steps.

| Method | Top-1 | Top-5 |
|---|---|---|
| Ours (w/o) adapter | 47.8 | 73.9 |
| Ours | **49.7** | **75.0** |

(c) Effect of temporal adapter.

**Effect of number of cached captions:** We evaluate the impact of $K$ in Alg. 1 (the number of generated captions cached per video). The results of Table 4a suggest that increasing $K$ from 1 to 3 offers a significant improvement in terms of Top-1 accuracy. However, going beyond $K = 3$ does not offer further gains, indicating that samples ranked lower are less likely to be representative of the video.

Table 3: Effect of retrieval step as measured on HMDB-51 for zero-shot recognition. We observe that retrieval is crucial for training a high quality model.

| RISE w/o retrieval | RISE |
|---|---|
| 40.1 | **49.7** |

**Effect of number of retrieval steps:** We evaluate the effect of the frequency of updating the pseudo-captions (l.10-13 of Algorithm 1). Table 4b shows that updating the captions results in better performance, showing that, over the course of training, they become more semantically meaningful. See supp. mat. for qualitative examples showing the pseudo-caption evolution across the retrieval steps.

**Effect of temporal adapter:** In Table 4c, we evaluate the effect of the video adapter of Sec. 3.2 by training a model that applies only temporal average pooling in the last layer.

**Discriminative properties of RISE:** To better understand the effect of RISE, in Fig. 2, we visualize the t-SNE plot of the text embeddings for the initial and final (after RISE training) pseudo-

captions. We observe that, without explicit guidance, RISE learns to produce more discriminative pseudo-captions. This showcases that RISE can be also interpreted as a self-supervised clustering mechanism. For further analysis, see appendix.

Table 5: Zero-shot classification results on HMDB-51 and UCF-101 in terms of top-1 accuracy.

| Method | HMDB-51 | UCF-101 |
|---|---|---|
| Discriminative approaches | | |
| ER-ZSAR (Chen & Huang, 2021) | $35.3 \pm 4.6$ | $51.8 \pm 2.9$ |
| MUFI (Qiu et al., 2021) | 31.0 | 60.9 |
| ActionCLIP (Wang et al., 2021) | $40.8 \pm 5.4$ | $58.3 \pm 3.4$ |
| ClipBert (Lei et al., 2021) | $21.4 \pm 1.0$ | $27.8 \pm 0.8$ |
| Frozen (Bain et al., 2021) | $27.8 \pm 0.3$ | $45.9 \pm 1.3$ |
| ViSET-96 (Doshi & Yilmaz, 2022) | 40.2 | 68.3 |
| BridgeFormer (Ge et al., 2022) | $37.7 \pm 1.2$ | $53.1 \pm 1.4$ |
| AURL (Pu et al., 2022) | 40.4 | 60.9 |
| RISE_101 (Lin et al., 2022a) | $41.1 \pm 3.7$ | $58.7 \pm 3.3$ |
| X-CLIP (Ni et al., 2022) | $44.6 \pm 5.2$ | $72 \pm 2.3$ |
| X-Florence (Ni et al., 2022) | $48.4 \pm 4.9$ | $73.2 \pm 4.2$ |
| Generative approaches | | |
| RISE (Ours) | $49.7 \pm 1.14$ | $69.1 \pm 0.62$ |
| RISE (GIT) (Ours) | $\mathbf{50.1 \pm 0.57}$ | $\mathbf{73.5 \pm 0.61}$ |

## 6.2 COMPARISON WITH STATE-OF-THE-ART

**Zero-shot evaluation on HMDB-51 & UCF-101:** Table 5 shows zero-shot classification results for the most standard benchmarks of HMDB-51 & UCF-101, where we also compare RISE against the current state-of-the-art. Despite the generative nature of our approach that was trained without any labels, our method sets a new state-of-the-art result on both HMDB-51 and UCF-101.

Table 6: Zero-shot classification results on Kinetics-220 (1-vs-160 setting).

| Method | Top-1 | Top-5 |
|---|---|---|
| Discriminative approaches | | |
| GCN (Ghosh et al., 2020) | $22.3 \pm 0.6$ | $49.7 \pm 0.6$ |
| ER-ZSAR (Chen & Huang, 2021) | $42.1 \pm 1.4$ | $73.1 \pm 0.3$ |
| X-CLIP (Ni et al., 2022) | $65.2 \pm 0.4$ | $86.1 \pm 0.8$ |
| X-Florence (Ni et al., 2022) | $\mathbf{68.8 \pm 0.9}$ | $\mathbf{88.4 \pm 0.6}$ |
| Generative approaches | | |
| RISE (Ours) | $51.7 \pm 1.1$ | $75.2 \pm 0.4$ |
| RISE (GIT) (Ours) | $53.0 \pm 1.2$ | $77.1 \pm 0.3$ |

Table 7: **Generalized** zero-shot classification results on Kinetics-220 (1-vs-620 setting).

| Method | Top-1 | Top-5 |
|---|---|---|
| Discriminative approaches | | |
| X-CLIP (Ni et al., 2022) | $14.76 \pm 0.51$ | $60.93 \pm 0.25$ |
| Generative approaches | | |
| RISE (Ours) | $29.51 \pm 0.71$ | $56.12 \pm 0.37$ |
| RISE (GIT) (Ours) | $\mathbf{31.20 \pm 0.64}$ | $\mathbf{58.04 \pm 0.41}$ |

**Zero-shot evaluation on Kinetics-220:** We report the results of RISE against state-of-the-art methods in Tables 6 and 7 under *two different scenarios*. In Table 6, we report the results obtained under

Table 8: Few-shot classification results on HMDB-51 and UCF-101 in terms of top-1 (%) accuracy.

| Method | Supervised dataset | HMDB-51 | | | | UCF-101 | | | |
|---|---|---|---|---|---|---|---|---|---|
| | | 2 | 4 | 8 | 16 | 2 | 4 | 8 | 16 |
| Discriminative approaches | | | | | | | | | |
| TimeSformer (Bertasius et al., 2021) | ✓ | 19.6 | 40.6 | 49.4 | 55.4 | 48.5 | 75.6 | 83.7 | 89.4 |
| Swin-B (Liu et al., 2022) | ✓ | 20.9 | 41.3 | 47.9 | 56.1 | 53.3 | 74.1 | 85.8 | 88.7 |
| X-CLIP (Ni et al., 2022) | ✓ | 53.0 | 57.3 | 62.8 | 64.0 | 76.4 | 83.4 | 88.3 | 91.4 |
| X-Florence (Ni et al., 2022) | ✓ | 51.6 | 57.8 | **64.1** | 64.2 | 84.0 | 88.5 | 92.5 | **94.8** |
| Generative approaches | | | | | | | | | |
| RISE (Ours) | ✗ | 54.0 | 59.1 | 62.1 | 64.0 | 88.2 | 90.2 | 92.6 | 93.5 |
| RISE (GIT) (Ours) | ✗ | **56.2** | **61.0** | **64.1** | **65.0** | **88.4** | **90.5** | **93.6** | **94.8** |

the standard setting of *novel, unseen classes* where the evaluation is done over the restricted set of novel classes only, i.e. where the models have to perform 1-vs-160 classification[2]. Under the standard zero-shot setting, our model is only surpassed by the very recent work of Ni et al. (2022) which is trained in a supervised manner on Kinetics-400. However, Table 7 shows the results for the more challenging *generalized zero-shot* setting, i.e. where the model is evaluated under a 1-vs-620 classification setting. For this setting, our method largely outperforms Ni et al. (2022). This shows that actually Ni et al. (2022) overfits the base classes of Kinetics-400, but for the standard zero-shot setting this is not "visible" as the base classes are excluded from evaluation. Overall, these results illustrate that our RISE is the best performing method on Kinetics-220, too.

**Zero-shot video captioning on Vatex:** While not trained for captioning, herein, we evaluate the performance of our approach for zero-shot video captioning on Vatex (Wang et al., 2019) in terms of CIDEr (Vedantam et al., 2015) score. As the results from Table 9 show, our approach largely matches the performance of Flamingo-3B and Flamingo-9B. Note that Flamingo models are (1) significantly larger (up to 20x for the 9B model), are trained using **2.1B** images and **27M** videos, and (3) use webly supervision. In contrast, our method is trained without any labels on Kinetics-400 (0.24M videos only).

Table 9: Zero-shot video captioning on Vatex.

| Method | CIDEr |
|---|---|
| Flamingo-3B (Alayrac et al., 2022) | **40.1** |
| Flamingo-9B (Alayrac et al., 2022) | 39.5 |
| RISE (Ours) | 39.6 |
| RISE (GIT) (Ours) | 40.0 |

**Few-shot evaluation on HMDB-51/UCF-101:** Results are reported in Table 8. Notably, our approach gets a large boost with minimum additional training, surpassing all prior methods and setting a new state-of-the-art. This showcases our model's ability to adapt to changes in vocabulary using few samples in an effective manner.

# 7 INSIGHTS, IMPACT & CONCLUSIONS

Our work is the first one to perform zero-shot action recognition using a generative model. Unlike discriminative models, our model's ability to generate free-form text has the following advantages: (a) it does not require a priori definition of the action classes, (b) it captures subtle class granularity, and (c) it produces interpretable/human-readable output. See appendix for more detailed analysis.

In contrast to discriminative models, we identify that generative models trained to predict directly the class names do not manifest zero-shot properties (Table 1). To address this, we propose to use free-form action pseudo-captions that are more fine-grained and generally explain lower level concepts that can be composed to describe higher level ones such as unseen actions (see appendix for qualitative results). Critically, to generate the pseudo-captions, we introduce RISE, a training framework consisting of two key components: (a) Self-training for pseudo-caption generation; and (b) Retrieval for enhancing the diversity of the pseudo-captions for each video. Finally, we note that without the retrieval module (i.e. using only the pseudo-labels of the current video) the accuracy is 20% lower.

---

[2]While there are a total of 220 unique classes in Kinetics-220 each split contains 160 only.

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

## A    GENERATIVE CLASSIFICATION ADVANTAGES

Our work establishes a new line of research, being the first one to perform zero-shot action recognition using a generative model. Unlike discriminative models, the generative model's ability to generate free-form text presents the following main advantages:

1. It is truly open-world as it does not require an a priori definition of the action classes. This allows the system to be deployed without explicit knowledge of the subdomain in question.

2. It is capable of capturing subtle class granularity not present within the target class, e.g.: "baby crawling" (original ground truth class name) vs "a baby crawling on the floor and looking at the camera" (ours). This is also notable by analyzing the size of the vocabulary used by the original Kinetics-400 labels vs ours: $522$ vs $14,140$ words. Moreover, within the user study performed, when correct, the human annotators preferred our predictions compared to the ground truth class name.

3. Our approach can provide alternative descriptions that explain the video using decoding techniques such as Beam Search or Nucleus sampling Holtzman et al. (2019). This naturally reflects the properties of real data, and human behavior in general, where the action description of a given sample might not unique.

## B    PSEUDO-CAPTION ANALYSIS BEFORE AND AFTER RISE TRAINING

**Pseudo-caption analysis before RISE:** The initial set of pseudo-captions is produced on a frame-by-frame basis using an image captioning model (i.e. BLIP). In Fig. 3, we analyze the number of unique pseudo-labels per video. Notice that, for most of the videos, there is no intra-agreement between frames (i.e. 2 or more frames have different pseud-captions). Overall, for the 241,000 training videos, initially, we have 316,616 unique pseudo-captions that contain 14,140 unique words. When analyzed on a per-class basis, each class contains between 200 and 3500 unique pseudo-captions. Fig. 4 illustrates this for a subset of classes (blue bars).

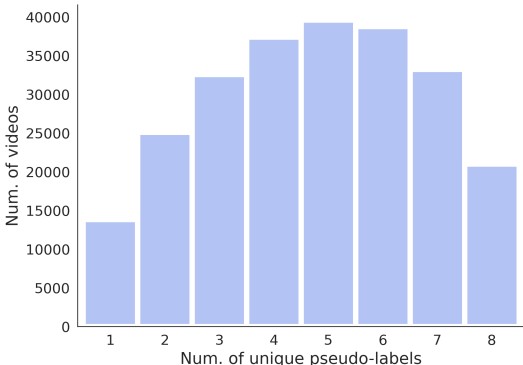

Figure 3: **Per video pseudo-caption diversity before RISE:** The histogram depicts the number of videos with $k \in [1, 8]$ unique pseudo-labels *prior to RISE training*. $k = 8$ indicates that all per frame pseudo-captions were initially unique, while $k = 1$ that all pseudo-captions were initially identical.

**Pseudo-caption analysis after RISE:** Herein, we study how the pseudo-labels change after RISE training. In terms of diversity, as shown in Fig. 4, the number of unique pseudo-caption decreases. This indicates that the RISE training filters out poor captions and consolidates the good ones. The later aspect is well-reflected in Fig. 5 that shows the percentage of videos whose pseudo-caption changed before and after RISE. Notice that, for the vast majority of the classes shown, more than 80% of the samples had their initial pseudo-caption changed. Moreover, the final pseudo-captions are more discriminative with respect to the class names. This is visualized in Fig. 6 where we show the cosine similarity between the pseudo-captions and the class names before and after RISE training. It can be observed that, without an explicit supervisory signal, RISE naturally improves their alignment to the expected classes.

## C  USER STUDY

We evaluate the capacity of CLIP-based top-k to correlate with human judgment by conducting a user study on 1,000 videos randomly selected from Kinetics-600. Following Levinboim et al. (2019), we formulate the correctness of a video-caption pair as the binomial probability $\hat{p} = P(CORRECT|video, caption)$ that can be estimated from the Bernoulli process. Each trial corresponds to a different human evaluator. The evaluators are shown an input video, which can be visualized multiple times, and are concomitantly asked the following question: *Does the text describe the action shown in the video?*. The raters are requested to choose between *yes* or *no*. To avoid inducing any bias, the interpretation of what represents a "correct" caption is left to the human evaluators to interpret and decide. While this could lead to unstable and inconsistent ratings, Levinboim et al. (2019) showed that with sufficient annotators (8-10) the results become stable and reproducible (we used 8). The final accuracy score is produced by taking the average of the binary annotations of each annotator on a per-sample basis and thresholding it to 0.5.

**Conclusion:** The results of the above experiment show that human annotators agree with the CLIP-based top-k metric in 75.67% of the cases, further validating the correctness of the latter to act as a proxy for classification. The accuracy measured by CLIP-based top-k on the Kinetics-600 subset used was 71.0% while the annotators' accuracy was 73.80%.

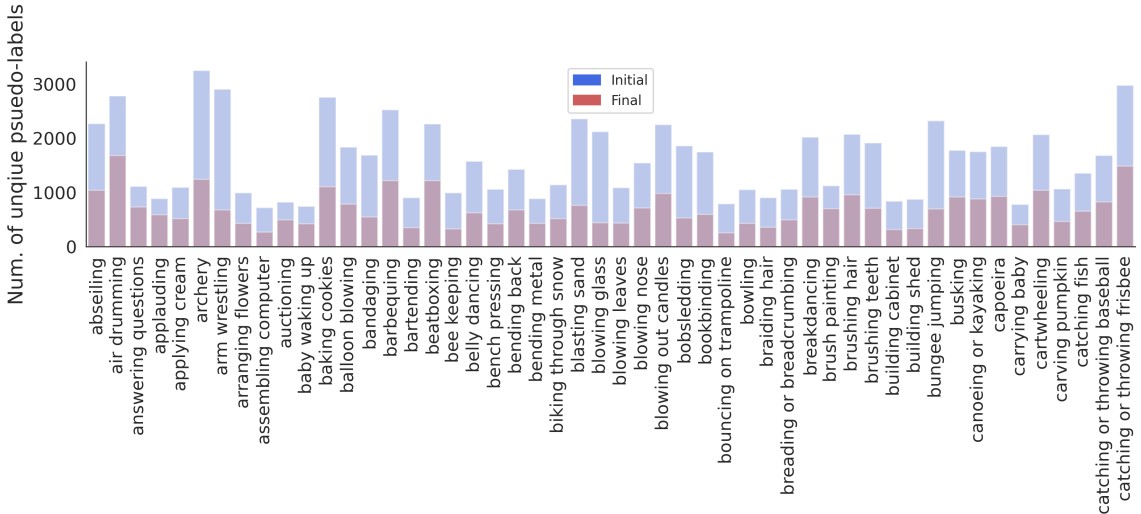

Figure 4: **Per class pseudo-caption diversity after RISE:** The number of unique pseudo-labels before RISE are shown in blue, and, respectively, at the end of RISE training are shown in red. Notice that our approach filters out poor pseudo-captions, converging towards a subset that better reflects the input video. To facilitate visualisation, only the first 50 classes are shown, but similar conclusions can be drawn for the rest too.

## D  AAR ARCHITECTURAL VISUALISATION

Fig. 7 depicts the overall architecture of the Generative Action Recognition network used in our work. The architecture largely follows the architecture proposed in BLIP Li et al. (2022). Note however that RISE is agnostic to the exact architecture. This was showcased in the paper by reporting experiments using GIT Wang et al. (2022).

## E  TRAINING HYPERPARAMETERS

They are listed in Table 10.

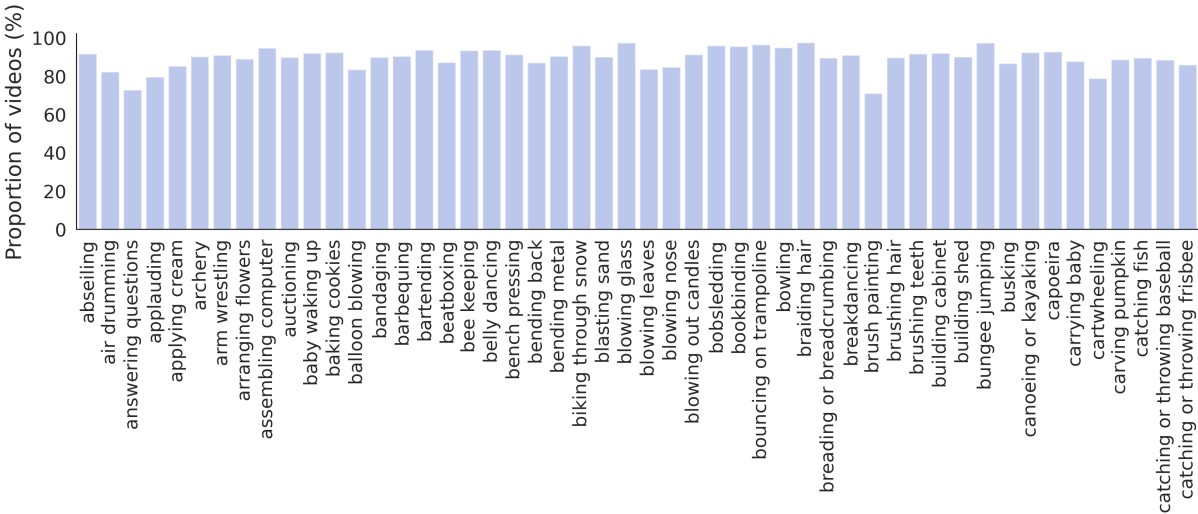

Figure 5: **Per-class proportion of videos that had their pseudo-caption changed after RISE**. To facilitate visualization, only the first 50 classes are shown, but similar conclusions can be drawn for the rest too.

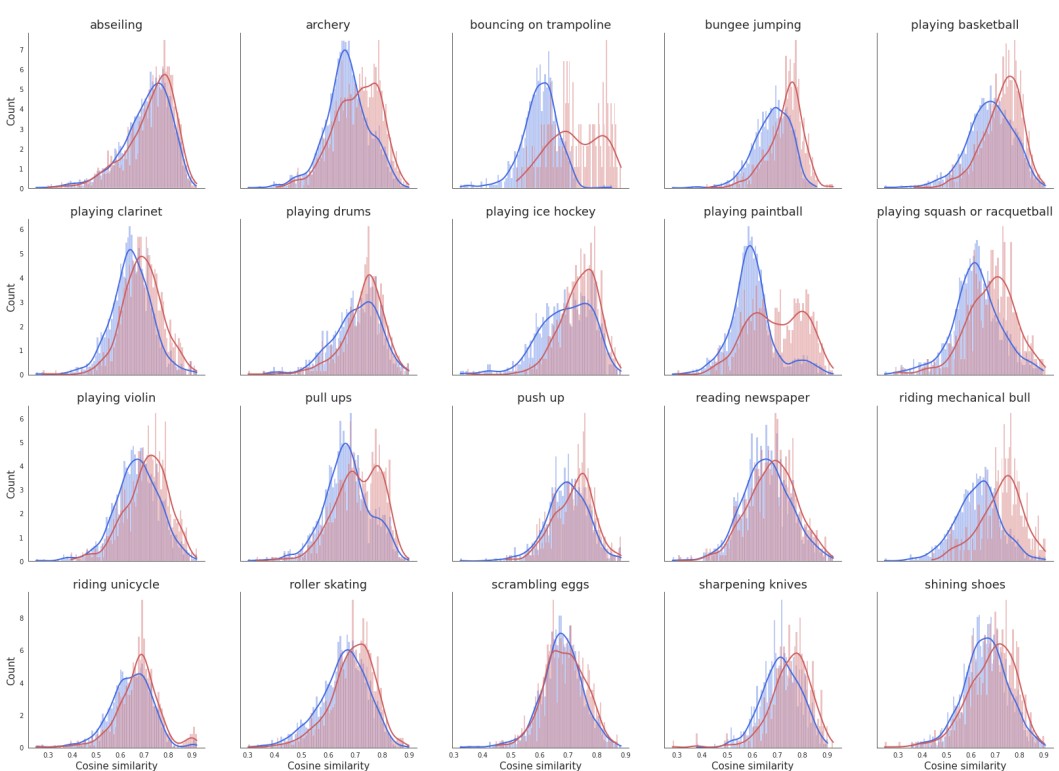

Figure 6: Cosine similarity between the class centroid and the predicted pseudo-labels for a random subset of 20 classes after pseudo-label initialization (blue) and at the end of the training using our approach (red). Notice that our approach, without any labels, improves upon the correctness of the initial predictions.

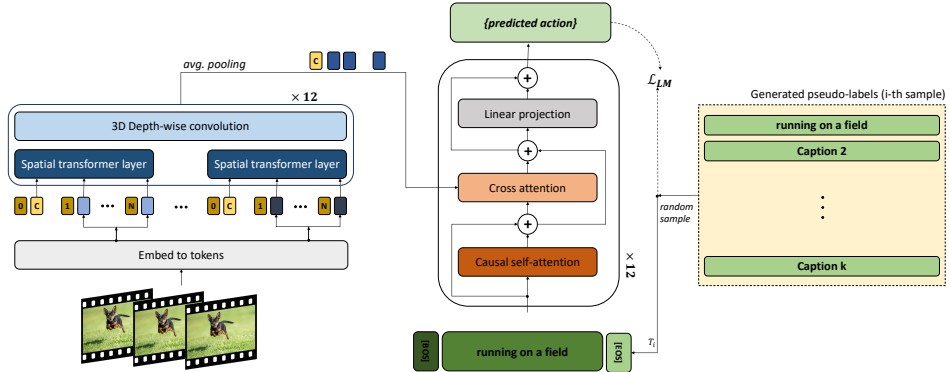

Figure 7: **Generative Action Recognition network.** We largely maintain BLIP's generative architecture Li et al. (2022) adding only a lightweight temporal adapter in a form of a 3D depth-wise convolution to combine information across frames.

|  | RISE training | Few-shot fine-tuning |
|---|---|---|
| Optimization | | |
| Optimizer | AdamW Loshchilov & Hutter (2017) | |
| Optimizer betas | (0.9, 0.98) | |
| Batch size | 64 | |
| Weight decay | 0.001 | |
| Learning rate scheduler | cosine | |
| Initial learning rate | $2e - 5$ | |
| Minimal learning rate | $2e - 8$ | |
| Epochs | 60 | max(400/K, 30) |
| Data augmentation | | |
| Random Flip | 0.5 | |
| Multi Scale Crop | (0.66, 0.75, 0.875, 1.0) | |
| Color Jitter | 0.8 | |
| Gray Scale | 0.2 | |
| Label smoothing | 0.2 | |

Table 10: Training hyperparameters. K is the number of samples per class for few-shot learning.

## F  ADDITIONAL COMPARISONS WITH STATE-OF-THE-ART

For completeness, in Table 11 and 12 we compare against an additional body of works for the task of zero shot action recognition.

## G  QUALITATIVE RESULTS

**Qualitative results produced by our method:** In Fig. 8 we show some pseudo-captions produced by the proposed model on videos taken from the validation set of Kinetics-600. Observe that the pseudo-captions generated describe the action taking place in the videos with contextual details. Moreover, in certain instances, our approach can capture more multiple events occurring in the input sample.

**Evolution of generated pseudo-caption:** Fig. 9 shows the evolution of the top-1 pseudo-caption after consecutive self-train & retrieval steps. Notice the gradual improvement in quality as the model learns to self-correct itself.

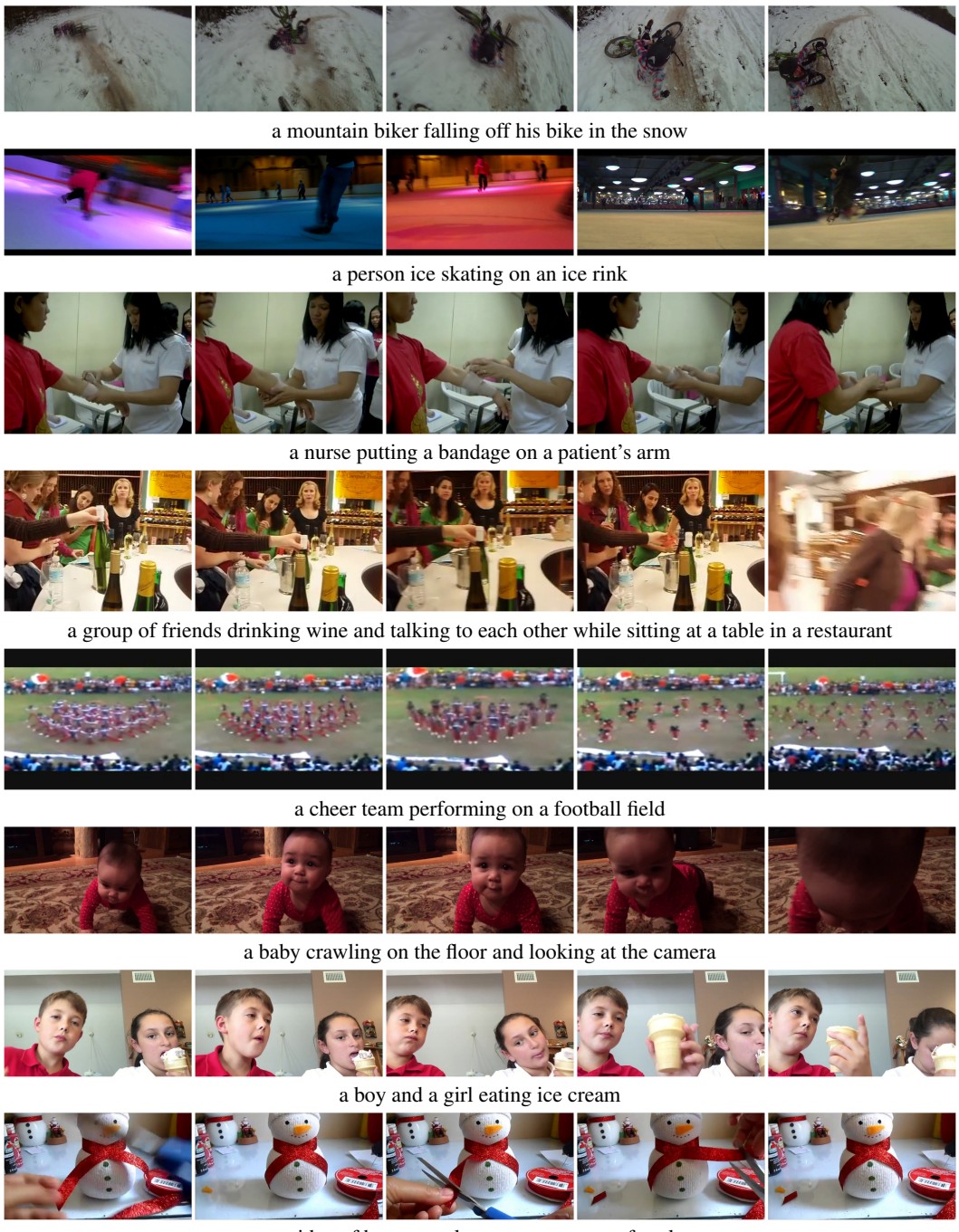

Figure 8: **Qualitative results:** Each row shows 5 frames sampled equidistantly from the video. Below the frames, we print out the predictions made by our model.

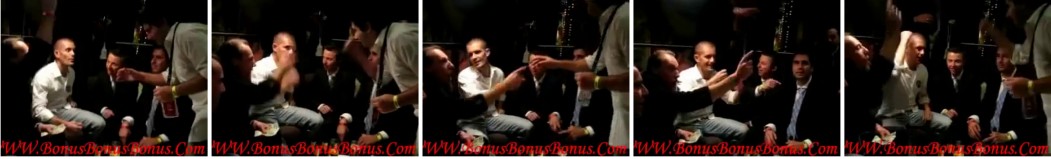

Step I: "a group of people standing around a table with a lit candle"
Step II: "a group of people standing around a table with a lit candle"
Step III: "a man and a woman playing a game of rock paper scissors"
Ground truth action (not used): "rock scissors paper"

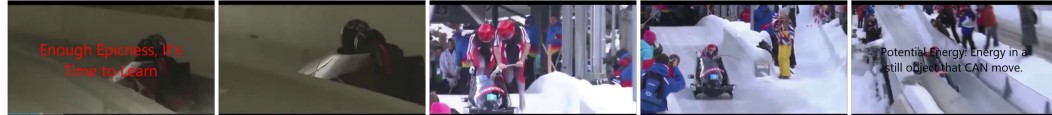

Step I: "a snowboarder falling off the side of a building"
Step II: "a bobsled bobsled race"
Step III: "a bobsleigh race"
Ground truth action (not used): "bobsledding"

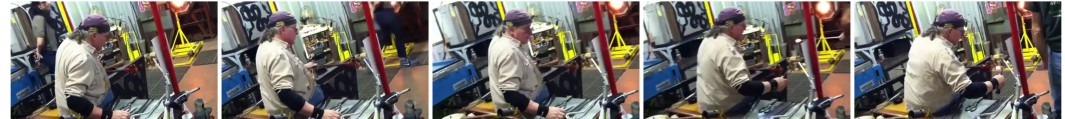

Step I: "a man in a hat and sunglasses working on a piece of metal"
Step II: "a glass blower blowing molten into the glass - blower stock videos and b - roll footage"
Step III: "a glass blower blowing glass"
Ground truth action (not used): "blowing glass"

Figure 9: **Qualitative results** showcasing the evolution of the most relevant pseudo-caption for each video after each retrieval step.

| Method | HMDB-51 | UCF-101 |
|---|---|---|
| Discriminative approaches | | |
| MTE Xu et al. (2016) | $19.7 \pm 1.6$ | $15.8 \pm 1.3$ |
| ASR Wang & Chen (2017) | $21.8 \pm 0.9$ | $24.4 \pm 1.0$ |
| ZSECOC Qin et al. (2017) | $22.6 \pm 1.2$ | $15.1 \pm 1.7$ |
| UR Zhu et al. (2018) | $24.4 \pm 1.6$ | $17.5 \pm 1.6$ |
| E2E Gao et al. (2019) | 32.7 | 48 |
| TS-GCN Brattoli et al. (2020) | $23.2 \pm 3.0$ | $34.2 \pm 3.1$ |
| ER-ZSAR Chen & Huang (2021) | $35.3 \pm 4.6$ | $51.8 \pm 2.9$ |
| MUFI Qiu et al. (2021) | 31.0 | 60.9 |
| ActionCLIP Wang et al. (2021) | $40.8 \pm 5.4$ | $58.3 \pm 3.4$ |
| ClipBert Lei et al. (2021) | $21.4 \pm 1.0$ | $27.8 \pm 0.8$ |
| Frozen Bain et al. (2021) | $27.8 \pm 0.3$ | $45.9 \pm 1.3$ |
| ViSET-96 Doshi & Yilmaz (2022) | 40.2 | 68.3 |
| BridgeFormer Ge et al. (2022) | $37.7 \pm 1.2$ | $53.1 \pm 1.4$ |
| AURL Pu et al. (2022) | 40.4 | 60.9 |
| ResT_101 Lin et al. (2022a) | $41.1 \pm 3.7$ | $58.7 \pm 3.3$ |
| X-CLIP Ni et al. (2022) | $44.6 \pm 5.2$ | $72 \pm 2.3$ |
| X-Florence Ni et al. (2022) | $48.4 \pm 4.9$ | $73.2 \pm 4.2$ |
| Generative approaches | | |
| RISE (Ours) | $49.7 \pm 1.14$ | $69.1 \pm 0.62$ |
| RISE (GIT) (Ours) | $\mathbf{50.1 \pm 0.57}$ | $\mathbf{73.5 \pm 0.61}$ |

Table 11: Zero-shot classification results on HMDB-51 and UCF-101 in terms of top-1 accuracy.

| Method | Top-1 | Top-5 |
|---|---|---|
| Discriminative approaches | | |
| DEVICE Frome et al. (2013) | $23.8 \pm 0.3$ | $51.0 \pm 0.6$ |
| ALE Akata et al. (2015a) | $23.4 \pm 0.8$ | $50.4 \pm 1.4$ |
| SJE Akata et al. (2015b) | $22.3 \pm 0.6$ | $48.2 \pm 0.4$ |
| ESZSL Romera-Paredes & Torr (2015) | $22.9 \pm 1.2$ | $48.3 \pm 0.8$ |
| DEM Zhang et al. (2017) | $23.6 \pm 0.6$ | $49.5 \pm 0.4$ |
| GCN Ghosh et al. (2020) | $22.3 \pm 0.6$ | $49.7 \pm 0.6$ |
| ER-ZSAR Chen & Huang (2021) | $42.1 \pm 1.4$ | $73.1 \pm 0.3$ |
| X-CLIP Ni et al. (2022) | $65.2 \pm 0.4$ | $86.1 \pm 0.8$ |
| X-Florence Ni et al. (2022) | $\mathbf{68.8 \pm 0.9}$ | $\mathbf{88.4 \pm 0.6}$ |
| Generative approaches | | |
| RISE (Ours) | $51.7 \pm 1.1$ | $75.2 \pm 0.4$ |
| RISE (GIT) (Ours) | $53.0 \pm 1.2$ | $77.1 \pm 0.3$ |

Table 12: Zero-shot classification results on Kinetics-220 (1-vs-160 setting).

