# OpenReview forum: "Unsupervised open-vocabulary action recognition with an autoregressive model"
_ICLR.cc/2024/Conference — Submitted to ICLR 2024_

### Official Review · Reviewer_o3GA · 2023-10-30

**Soundness:** 3 good
**Presentation:** 3 good
**Contribution:** 4 excellent
**Rating:** 8
**Confidence:** 4

**Summary:**

The research introduces a novel approach to zero-shot action recognition by adapting a pre-trained autoregressive Vision & Language model to generate action-specific video captions. Instead of using conventional contrastive learning, the authors implement an unsupervised learning framework. This framework combines self-training with pseudo-caption generation and a retrieval system to source diverse pseudo-captions for each video, enhancing the training dataset's variety. Trained model has the ability to create predictions that are detailed, interpretable, and inherently open-vocabulary. The resulting approach has demonstrated proficiency in zero- and few-shot action recognition scenarios, either equating or surpassing the performance of prevailing contrastive learning methods.

**Strengths:**

1. Innovative Research Direction: The paper's proposition to generate free-form text labels for video clips aligns with the evolving trajectory of Vision-Language Models (VLM). I believe event understanding essentially boils down to crafting a concise yet descriptive label for a given video clip. This approach not only offers a semantically interpretable compression of the clip but also ensures that the textual representation is more compact than raw pixel sequences. This compressed representation potentially facilitates deeper comprehension of longer videos. Given this context, the conventional use of pre-defined class labels seems restrictive. The generative approach showcased in this paper appears to be a more optimal strategy. If this is truly the "first of its kind," as the authors suggest (I admit, my expertise in this domain might not be exhaustive), it could mark a significant paradigm shift in the field.


2. Rigorous Experimental Validation: The authors have meticulously demonstrated the significance of each component of their proposed methodology. Their ablation studies robustly highlight the indispensable nature of each segment for achieving commendable results. The comparisons made with existing works are also comprehensive and fair.

**Weaknesses:**

Presentation of Methodology: My primary concern lies in the method section's exposition. While the core idea of the proposed method is intuitive and straightforward, the section seems overburdened with intricate notations. This makes it somewhat challenging to follow. A more streamlined and intuitive presentation could significantly enhance the paper's readability and impact.

**Questions:**

1. While the author mentions that video-video similarity scores, leveraging the frozen CLIP encoder, can be pre-computed and re-utilized throughout training, Algorithm 1, line 12, indicates these scores are calculated during training. Could you clarify this discrepancy?


2. Regarding the CLIP image encoder's usage for retrieval, is there any temporal modeling involved? Is the process simply averaging the frame-wise representation?


3. Is it feasible to concurrently train the video retrieval model?

---

> ### Author Response · Authors · 2023-11-19
> **Reply to Reviewer o3GA**
>
> We appreciate the reviewers comments, and we thank them for their time and considerations.
>
> __Q1__: _"Presentation of Methodology: My primary concern lies in the method section's exposition. While the core idea of the proposed method is intuitive and straightforward, the section seems overburdened with intricate notations. This makes it somewhat challenging to follow. A more streamlined and intuitive presentation could significantly enhance the paper's readability and impact."_
>
> __R1__: Thank you for your feedback. We will improve the clarity of the exposition further by the camera ready.
>
> __Q2__: _"While the author mentions that video-video similarity scores, leveraging the frozen CLIP encoder, can be pre-computed and re-utilized throughout training, Algorithm 1, line 12, indicates these scores are calculated during training. Could you clarify this discrepancy?"_
>
> __R2__: Line 12 is executed every R-epochs, when the pseudo-labels are reassigned. You are correct, as only the text changes, the remaining features and scores that don't change (i.e. video-to-video) are cached in practice during the 1st run. We will clarify this in the text.
>
> __Q3__: _"Regarding the CLIP image encoder's usage for retrieval, is there any temporal modeling involved? Is the process simply averaging the frame-wise representation?"_
>
> __R3__: Yes, that's correct. For the type of videos used (which are 3-15 seconds) this isn't generally a problem. We explored multiple avenues on this (e.g.: aggregated scoring on a frame-by-frame basis, using X-CLIP video encoder etc), but found the performance to be similar.
>
> __Q4__: _"Is it feasible to concurrently train the video retrieval model?"_
>
> __R4__: Thank you for your question and suggestion. We believe that it’s possible to train a retrieval component too, following BLIP modeling for example, where the LM is used both as an encoder and decoder. In this work, we only focused however on the generative component, adapting it for action recognition.

---

> ### Comment · Reviewer_o3GA · 2023-11-22
> **Thank you for the reply!**
>
> After reading other reviewers' comments and authors' responses, I decided to keep my original rating. But the presentation quality is not a trivial issue. (note that the clarity problem is mentioned by almost all reviewers) I hope the authors improve the presentation quality in their revision.

---

> > ### Author Response · Authors · 2023-11-22
> > **Thank you**
> >
> > Thank you! We will of course take on board all the suggestions for improving the structure and presentation of the paper when preparing the revised version of the manuscript.

---

### Official Review · Reviewer_RvG7 · 2023-10-31

**Soundness:** 3 good
**Presentation:** 1 poor
**Contribution:** 3 good
**Rating:** 5
**Confidence:** 3

**Summary:**

The paper focuses on a method for zero-shot action recognition based on autoregressive Vision & Language (V&L) models and suggests a new training framework that combines retrieval and self-training (RISE)
This approach uses pseudo-captions generated from the model itself, enhances the diversity of these pseudo-captions with a retrieval module based on CLIP, and shows competitive or better results compared to existing methods.

**Strengths:**

This study pioneers the development of a generative V&L FM for open-vocabulary video recognition, marking a significant stride for the community.

Notably, the self-training procedure for REST requires no action-specific labels, allowing for RISE to be trained in a unsupervised fashion.

The efficacy of the proposed video-text retrieval is proved by the extensive experimental results.

**Weaknesses:**

While the paper offers valuable contributions, oversights such as typos, inconsistency, and table misalignments detract from its overall quality and can be distracting for readers. Maintaining consistent formatting throughout the paper can greatly enhance its quality.

- The method is primarily described in prose, making it somewhat challenging to grasp. Incorporating relevant figures could make the content more accessible and easier to follow.
- In Section 5's second paragraph, avoid using abbreviations for "resolution."
- The layouts of Tables 3 and 4 could benefit from better alignment.
- For Table 2, the emphasis (bold) should be on 50.1. Row entries should maintain consistency, e.g., "RISE(ours)."
- In Table 4 (b), I guess the value 49.8 of the column for "$N_I=4$" is a typo since the best result is reported as 49.7, and the bold formatting for 49.8 should be removed. it may cause misunderstanding of the experiment result.
- The last sentence of section 3.2 uses the format "$\mathbb{R}^{(H \cdot W + 1)\times d}$." For uniformity, it should be revised to "$\mathbb{R}^{(H \times W + 1)\times d}$."
- In Table 8, the partition line should begin at the number 2

Reproducibility:

The code has not been provided. Given the complexity of the method, it would be beneficial for reproducibility if the authors could supply the code for review.

---------Post rebuttal---------
Having read the comments from other reviewers, I continue to lean towards rejecting this paper. I remain convinced that the quality of presentation is a fundamental aspect of any paper, and in this case, it falls below the standard required for acceptance.

**Questions:**

Would the authors be open to releasing the supporting code for their paper?

As far as I understand table 3 can be merged to table 4 as $N_I= 0$, do I understand correctly? and if the result of table 4-(b) is correct, is there any reason of not reporting the value with the better result of 49.8?

---

> ### Author Response · Authors · 2023-11-19
> **Reply to Reviewer RvG7**
>
> We thank the reviewer for the provided comments and for acknowledging our "valuable contribution". We are surprised however by the low score given for what are mostly cosmetic issues, which we believe can be easily addressed by the camera ready. Furthermore, we hope that the reviewer will reassess their scoring.
>
> __Q1-7__: _"The method is primarily described in prose, making it somewhat challenging to grasp. Incorporating relevant figures could make the content more accessible and easier to follow." + others comments regarding cosmetic issues: e.g: the alignment of the tables, typos, partition line etc."_
>
> __R1-7__: We will improve this further, incorporating your suggestions. We note however that Fig. 1 describes the overall framework at a high level, while Alg. 1 already offers an exact description of how our method works. We will also address all the remaining cosmetic issues, carefully proofreading the paper.
>
> __Q8__: _"The code has not been provided. Given the complexity of the method, it would be beneficial for reproducibility if the authors could supply the code for review."_
>
> __R8__: Thank you, we aim to release it upon acceptance.
>
> __Q9__: _"As far as I understand table 3 can be merged to table 4 as , do I understand correctly? and if the result of table 4-(b) is correct, is there any reason of not reporting the value with the better result of 49.8?"_
>
> __R9__: We believe that keeping them separate could make it easier to follow in the paper, when read together with the sections that reference them. We report 49.7 as this is the model we used in the end. We didn't consider the 0.1% sufficiently large to justify the additional number of epochs. The results reported for 4 steps is simply to showcase that the performance saturates afterward.

---

### Official Review · Reviewer_JWgN · 2023-11-01

**Soundness:** 2 fair
**Presentation:** 1 poor
**Contribution:** 2 fair
**Rating:** 5
**Confidence:** 3

**Summary:**

This paper tackles the problem of zero-shot action recognition using generative models. Given a pretrained vision and language model that could generate captions, the proposed method utilizes CLIP and a retrieval-based self-training paradigm to improve the model without additional action labels. Experiments on three zero-shot and few-shot action recognition benchmarks show the efficacy of the proposed method.

**Strengths:**

1. The proposed method is new for zero-shot action recognition.
2. The generative approach in theory is more flexible than the discriminative methods and it can facilitate the open-vocabulary setting.

**Weaknesses:**

1. The performance gap between the proposed method and discriminative methods with similar pretraining models (like X-CLIP) is huge (on the standard zero-shot benchmark, 65.2 vs. 53.0, Table 6).
2. The paper writing/clarity needs to be greatly improved:
2.1 The overview figure (Figure 1) is not clear. There is no H, g_v, g_t, language modeling loss, etc. in the figure.
2.2 In Table 2, “Dataset” should be “Pretrained Dataset” and for the RISE row shouldn’t it be “LAION-115M & Kinetics-400” since it is initialized from BLIP?
2.3 Minor presentation issues:
What is FM (Line 1)? Foundation model? This has not been specified before.
H \dot W + 1 or H \times W + 1? Be consistent.

**Questions:**

1. Inference speed comparison. I’m curious about the inference speed difference between the discriminative and the proposed generative approach.
2. I think “zero-shot” and “open-vocabulary” are not entirely the same in terms of experimental evaluation. One can claim zero-shot ability on a limited set of classes but for open-vocab one needs to build a much larger test than 600 classes. More discussion on this is needed or the authors could avoid claiming “open-vocabulary”.

-----Post-rebuttal comments: After reading other comments, I also have doubts about the novelty of this paper. The presentation of this paper is poor. The authors use zero-shot benchmarks for evaluation but claim "open-vocabulary". Therefore I am maintaining my score of leaning rejection.

---

> ### Author Response · Authors · 2023-11-19
> **Reply to Reviewer JWgN**
>
> We thank the reviewer for their time and comments. We hope to have addressed your remaining concerns below.
>
> __Q1__: _"The performance gap between the proposed method and discriminative methods with similar pretraining models (like X-CLIP) is huge (on the standard zero-shot benchmark, 65.2 vs. 53.0, Table 6)."_
>
> __R1__: We respectfully disagree with this assessment. This is cherry-picking from our results and is not representative of the overall performance achieved by our approach. This is the only case where our approach is outperformed by X-CLIP. __We outperform all prior works for zero-shot on HMDB-51, UCF-101 and K220 (1-vs-620 setting) and for few-shot recognition on UCF-101 and HDMB-51 - which are all the settings reported on X-CLIP too__. Note that we outperform it despite the fact that our approach uses pseudo-labels while X-CLIP for example, uses human-annotated labels.
>
> __Q2__: "The paper writing/clarity needs to be greatly improved: 2.1 The overview figure (Figure 1) is not clear. There is no H, g_v, g_t, language modeling loss, etc. in the figure. 2.2 In Table 2, “Dataset” should be “Pretrained Dataset” and for the RISE row shouldn’t it be “LAION-115M & Kinetics-400” since it is initialized from BLIP? 2.3 Minor presentation issues: What is FM (Line 1)? Foundation model? This has not been specified before. H \dot W + 1 or H \times W + 1? Be consistent."
>
> __R2__: Fig. 1 was intended as a high-level overview. The exact details can be followed in Alg. 1. We will add where possible such annotations to improve its readability.
>
> Regarding Table 2 - As the setting from Table 2 is for zero-shot, the training and pretraining are the same step. We will rename it for further clarity.
>
> Regarding whether, it should state LAION-115M & K-400, we included only the last dataset used for brevity. For example, BLIP vision encoder is pretrained on Imagenet, while their LM is a pretrained BERT, but they are usually not mentioned explicitly. We will add a note mentioning that the initial checkpoint is a BLIP model, hence was pretrained on the data from BLIP.
>
> Yes, FM stands for foundation model. We will introduce this.
>
> Regarding H \cdot W + 1, this was a typo. We will fix it.
>
>
>  __Q3__: _"Inference speed comparison. I’m curious about the inference speed difference between the discriminative and the proposed generative approach."_
>
> __R3__: Our generative approach consists of a vision encoder and text decoder, while for the discriminative case the model will typical encompass a vision and a text encoder (e.g. CLIP, X-CLIP, ActionCLIP, etc.). For clarity, we benchmarked each component independently. On our system, for 8 frames, for ours: vision encoder: 0.059s + text decoder: 0.012s; For an equivalent discriminative model: vision encoder: 0.058s + text encoder: 0.008 sec (x num classes);
> Notice that the models are comparable in terms of speed. Generally, the time will be dominated by the vision encoder, especially as the number of frames grows.
>
> __Q4__: _"I think “zero-shot” and “open-vocabulary” are not entirely the same in terms of experimental evaluation. One can claim zero-shot ability on a limited set of classes but for open-vocab one needs to build a much larger test than 600 classes. More discussion on this is needed or the authors could avoid claiming “open-vocabulary”."_
>
> __R4__: We agree in principle, however, there are no datasets that could appropriately measure this, and collecting one is a major endeavor and out of the scope of this work. We believe however generative models, as proposed in this work, are a natural fit for this, as they are not limited to a selection from a predefined set of labels. In a sense, our user study offers a glimpse of this, as the users are asked to assess whether a text produced describes the action or not, but as we used the K220 data, this will still not increase the variability of the classes themselves. We will add this discussion into the updated manuscript, addressing also the use of the term.

---

### Official Review · Reviewer_bK27 · 2023-11-01

**Soundness:** 2 fair
**Presentation:** 2 fair
**Contribution:** 2 fair
**Rating:** 3
**Confidence:** 5

**Summary:**

The paper proposes an approach to solve the problem of open-vocabulary action recognition using a generative auto-regressive model. The approach includes two modules: 1) Self-training using model generated pseudo-captions and no labels for open-world settings and 2) CLIP based Retrieval to introduce diversity in the pseudo-captions.

**Strengths:**

-	The proposed approach shows good results on multiple datasets.

**Weaknesses:**

-	The novelty of the proposed approach is limited. Basically, it is a mixture of BLIP and CLIP iterating over multiple times throughout the training. CLIP will generate new text features because of new text generated by the updated BLIP model after each iteration. It looks like the model is finetuned on the Action Recognition dataset.
-	(Section 1) Table-1 - Are the approaches trained from scratch or fine-tuned? How is the testing done? What is zero-shot accuracy without any fine-tuning at all? Because in Section 3.1 it is mentioned that the model is initialized with pre-trained weights.
-	(Algorithm/Training Efficiency) -
-	From the algorithm it looks like the video-video and video-text similarity scores are calculated after every training iteration. Given the dataset size, how is it not a bottleneck, because it will be a huge operation to generate captions for all training videos and then calculate similarity between all data-points? (Lines 10 and 12).
-	Why is it even needed to calculate video similarity after each iteration? It won’t change as CLIP encoders are frozen.
-	(Section-3.4) Similarity Calculation - Why not similarity be calculated using CLIP models adapted for video such as [1]? They are better encoders in the case of videos.
-	(Section 3.5) Retrieval - How does the proposed approach ensure that the captions selected each time introduce diversity via CLIP? Because the video-video similarity is calculated from a frozen encoder.  The video will be the same, only the text will be different for that same video because BLIP is finetuned on K400 now.
-	General Concern - BLIP/BLIP-2 is supposed to be made for tasks like image-retrieval which means it has a better capability to find the images given the phrases. Does this setting is more like a video-text retrieval task rather than solving action recognition? Why is it mentioned again and again there are no labels? If I generate a text for UCF101 class from BLIP for applying eye makeup - BLIP generates multiple texts for a woman putting on makeup. Now if we compare the similarity of this phrase with classes of UCF-101 it will be sure that the ApplyEyeMake up class will have high scores at the inference time. I personally tested it with Huggingface APIs.
-	(Section 6.1) Ablation study:
-	(Table-2) Comparing an image model against a video model is not the right thing to do. For fair comparison, it should have been a BLIP model with adapter but no other additions. Only trained with class labels/original captions.


[1] Rasheed, Hanoona et al. “Fine-tuned CLIP Models are Efficient Video Learners.” 2023 IEEE/CVF Conference on Computer Vision and Pattern Recognition (CVPR) (2022): 6545-6554.


---------

POST REBUTTAL

---------

Thanks to the authors for the clarifications; the authors addressed some of my concerns. However, the following issues are still there:

1.	Oversimplification - I don’t think using complex terms to present a simple approach could be mentioned as novelty and that’s why I kept the presentation score low. There’s very limited novelty to this work. It’s just usage of the CLIP and BLIP image-based approach and there’s no adaptation for Videos. An action recognition task involves temporal understanding. I don’t see any novel component in the proposed approach, quantitatively or qualitatively which shows the temporal understanding. The reason why I’m saying HMDB and UCF couldn’t quantify temporal understanding is because if an action class is considered, the caption will always contain the original class description such as trampoline, cricket, etc. pick any frame from the video. Something-something will actually show that it focuses on the verb aspect which is considered as action. With the current approach, there’s no novelty in terms of temporal understanding for the task which is the basis of video action understanding.

2.	Looking at the second contribution, caption diversity selection. I don’t think it’s the CLIP which is selecting a better caption  (CLIP based retrieval), it’s just the better caption generated by BLIP as training progresses. There’s no criteria to improve the selection other than similarity scores which depend on text generated by BLIP. Both of the contributions are pretty weak at this point of time, that’s why I believe the paper needs more work.

3.	Ablation on UCF-101 - If we look at Table 4 a and b and Table 5, HMDB-51 scores match, however, the ablation on both K and N shows a drop of 7% on UCF-101? Why would that happen?

4.	Computation Aspect - If we look at Appendix B, 316k+ unique pseudo caption generation for 241K videos is not normal. It’s repeated at least three times as mentioned, training is broken down to 3 steps. Then, similarity is calculated. What’s the matrix size for similarity calculation? There should be a dedicated section about it in discussion/ablation - why wouldn’t this step increase computation? Right now, numbers are very huge and how it’s trimmed is not clear. Top-k similar videos top-k similar captions.

In the current state, I can’t change my rating. The paper requires following adaptations to make it better - a dedicated novelty to show adaptation for action recognition, qualitative visualization/quantitative analysis of temporal understanding,  filter of captions for better selection at each training stage, and a clear study on computational aspect given the size of dataset. The paper presentation also needs work.

**Questions:**

-	(Table 4-b) Why is it shown only for 4 iterations? The authors did 6 iterations. 60 epochs and each iteration are done after 10 epochs.
-	Does the model have any temporal understanding? Result on Something-Something v2 would have highlighted these capabilities.
-	Ablations on UCF even in the appendix would have been more convincing. HMDB is a small dataset.
-	Why not better backbone architectures such as BLIP-2 is used for the experiments?

---

> ### Author Response · Authors · 2023-11-19
> **Reply to Reviewer bK27 (part 1)**
>
> __Q1__: _"The novelty of the proposed approach is limited. Basically, it is a mixture of BLIP and CLIP iterating over multiple times throughout the training. CLIP will generate new text features because of new text generated by the updated BLIP model after each iteration. It looks like the model is finetuned on the Action Recognition dataset."_
>
> __R1__: We politely disagree, this is a misrepresentation of our work.
>
> Firstly, our work is the first to explore the idea of generative V&L models for open world action recognition - a novel concept and idea that was not previously proposed nor investigated, and we show it can match and even outperform discriminative methods without making use of human annotations/labels.
>
> Secondly, the description is both inaccurate and an oversimplification. Our process consists of a bi-level retrieval: video-to-video and  frame-to-text, which retrieves the appropriate pseudo-captions at a dataset level. We empirically show the retrieval module, which is non-trivial, has a massive impact in the final performance (+9%, see Table 3).  There are also no new text features computed after each iteration, they are only recomputed after a given number of epochs.
>
> Finally, no human annotated labels were used for training.
>
> __Q2__: _"Table-1 - Are the approaches trained from scratch or fine-tuned? How is the testing done? What is zero-shot accuracy without any fine-tuning at all? Because in Section 3.1 it is mentioned that the model is initialized with pre-trained weights._"
>
> __R2:__ In Table 1 all approaches are fine-tuned on Kinetics-400 using various forms of supervision (each row specifies the type of supervision and training strategy).  We already report the performance of the initial BLIP/GIT pretrained model (i.e. without any finetuning) in Table 2 on the HMDB-51 dataset. For K220, the performance of such a model without fine-tuning is ~10%. Our approach is massively better than this. The initial models are generally not sufficiently discriminative to form class-separable text. In fact, this is one aspect that our bi-level retrieval seeks to address.
>
> __Q3__: _"From the algorithm it looks like the video-video and video-text similarity scores are calculated after every training iteration. Given the dataset size, how is it not a bottleneck, because it will be a huge operation to generate captions for all training videos and then calculate similarity between all data-points? (Lines 10 and 12). Why is it even needed to calculate video similarity after each iteration? It won’t change as CLIP encoders are frozen."_
>
> __R3__: We don't do this, this is a misunderstanding. Firstly, as shown in Alg. 1 this is repeated after R epochs, not every iteration as stated. Secondly, the video-video similarity scores and the video features are cached, as they don’t change during training. Thirdly, as shown in Alg. 1 the text-to-video similarities are computed only between the captions of the videos that are close in the video-video space. All in all, given that most of the data is cached, the process is repeated after a few epochs, and it considers only similarities within a neighborhood, the process is computationally feasible, taking a similar amount of time with that of running one training epoch.
>
> __Q4__: _"Similarity Calculation - Why not similarity be calculated using CLIP models adapted for video such as [1]? They are better encoders in the case of videos."_
>
> __R4__: Our method is amenable to any vision-language model. We used a CLIP model simply because at the time it was by far the most widely used foundation model V&L, with well studied generalization properties. Thanks for pointing [1], it is relatively recent (CVPR’23) and we weren't aware of it at the time of developing our approach. Given that we already achieve state-of-the-art results, having scope for further improvements is actually not a negative but a good thing.
>
>  __Q5__: _"How does the proposed approach ensure that the captions selected each time introduce diversity via CLIP? Because the video-video similarity is calculated from a frozen encoder. The video will be the same, only the text will be different for that same video because BLIP is finetuned on K400 now."_
>
> __R5__: We are always maintaining a large pool of options, searching during the retrieval within the captions of other similar videos too. This ensures a high initial diversity. Such collapse can occur in practice only if the initial diversity is very low (not the case here) or if the model used for retrieval is inaccurate (i.e. unsuitable to match a sentence with a frame) - again not the case here. Our results and statistics about the data in the appendix support this assessment. We also show excellent experimental performance, which clearly would not be the case if the retrieval mechanism failed to maintain or improve diversity.

---

> ### Author Response · Authors · 2023-11-19
> **Reply to Reviewer bK27 (part 2)**
>
> __Q6__: _"General Concern. BLIP/BLIP-2 is supposed to be made for tasks like image-retrieval which means it has a better capability to find the images given the phrases. Does this setting is more like a video-text retrieval task rather than solving action recognition?"_
>
> __R6__: No, this is a classification problem, not a retrieval problem. Our solution is generative, i.e. it produces an action-descriptive output in an autoregressive manner. This is notably different from all prior action recognition methods."
>
> __Q7__: _"Why is it mentioned again and again there are no labels? If I generate a text for UCF101 class from BLIP for applying eye makeup - BLIP generates multiple texts for a woman putting on makeup. Now if we compare the similarity of this phrase with classes of UCF-101 it will be sure that the ApplyEyeMake up class will have high scores at the inference time. I personally tested it with Huggingface APIs."_
>
> __R7__: It is mentioned because our method, unlike the ones we compare with, doesn't use human-annotated labels, just pseudo-labels. What the reviewer is describing is exactly a pseudo-label. We aren't sure why there is a confusion here.
>
> While the outputs produced are generally relevant, they are not sufficiently discriminative for action recognition. In an action recognition setting, the predicted text must also be separable according to a given class list and specific/descriptive of the action. We provide quantitative results measuring exactly this initial performance: On large datasets, such as K-220 (1-vs-620 setting) the performance is very poor (around 10% as mentioned in our reply above, and as shown in the paper in Table 2 for HMDB-51). Please also note that in practice it's not known beforehand if the action is present and clear in a given frame, as in the example mentioned. Moreover, the image-based models will struggle with actions that require analyzing multiple frames at once.
>
> __Q8__: _"(Table-2) Comparing an image model against a video model is not the right thing to do. For fair comparison, it should have been a BLIP model with adapter but no other additions. Only trained with class labels/original captions."_
>
> __R8__: Please see below the results of the requested experiments:
>
> Training a generative model using class names results in near-complete overfitting to the base class names. This is one of the findings of our paper, shown in Table 1. Adding temporal adapters doesn't change the performance and such models score close to 0% top-1 accuracy (e.g. K220 1-vs-620 setting). For fine-tuning with the initial captions, without our approach, on HMDB the results are as follows: w\o temp. Adapter: 40.1%; with temporal adapter: 42.6%; Ours: 49.7%; Our approach is significantly better.
>
> __Q9__: _"(Table 4-b) Why is it shown only for 4 iterations? The authors did 6 iterations. 60 epochs and each iteration are done after 10 epochs."_
>
> __R9__: We use 3, not 4 for the final model. The 4-th is shown to showcase that performance saturates after, so they are not run. The 1st training round is longer (30 epochs), hence the total of 60.
>
> __Q10__: _"Does the model have any temporal understanding? Result on Something-Something v2 would have highlighted these capabilities."_
>
> __R10__: We follow the evaluation protocol and datasets from prior work (i.e. X-CLIP). SS-v2 is generally not a good fit for zero-shot based on text due to the unusually worded classes (e.g. : "Pretending to put [something] onto [something]"). We validate this already on HMDB-51 which has already a temporal nature (compared with UCF-101).
>
> __Q11__: _"Ablations on UCF even in the appendix would have been more convincing. HMDB is a small dataset."_
>
> __R11__: We are already reporting results on UCF-101, HMDB-51, K-620 (2 settings). We don't see why the conclusions would change for the ablations when measured on UCF-101. For prior methods, generally, these performances correlate. Nevertheless, we followed your suggestion:
>
> For different values of K: K=1 (62.6); K=3 (63.93); K=5 (63.98);
>
> For different number of retrieval steps: N=1 (59.6); N=2 (62.75); N=3 (63.93); N=4 (63.95).
>
> These results are in line with the ones for HMDB-51.
>
> __Q12__: _"Why not better backbone architectures such as BLIP-2 is used for the experiments?_"
>
> __R12__: We are already experimenting with two major architectures: BLIP and GIT. We show that better models (i.e. GIT) lead, as expected, to better results. Due to resource limitations, we kept our experiments contained to more computationally affordable model sizes, hence didn’t experiment with more methods or even larger models as in BLIP-2. We believe the current experiments already showcase that our approach is generic and can benefit from better models. There will always be a larger model we could try.

---

### Meta-Review · Area_Chair_isVy · 2023-12-09

**Metareview:**

This work addresses zero-shot(? see note below) action recognition by generating captions for the given video, leveraging CLIP and BLIP.

The reviews are mixed - three of the four reviewers recommend reject or borderline reject (x2); one reviewer supports the work and recommends accept.  Having read through the reviews and responses, the AC agrees with the first reviewer that this paper is not ready for publication.  There are to many remaining concerns regarding the contribution, justification for various design choices and overall paper presentation.

====

There seems to be some confusion on the distinction between open-vocabulary vs. zero-shot and the two terms and settings are used interchangeably.  In the zero-shot case, the classes are known during training (even if there are no training samples) vs. open-vocabulary where classes are not known during training.

**Justification For Why Not Higher Score:**

clarity, justification and presentation issues

**Justification For Why Not Lower Score:**

N/A

---

### Decision · Program_Chairs · 2024-01-16

Reject